# Integrative Proteomic and Phosphoproteomic Analyses of Hypoxia-Treated Pulmonary Artery Smooth Muscle Cells

**DOI:** 10.3390/proteomes10030023

**Published:** 2022-06-28

**Authors:** Ang Luo, Rongrong Hao, Xia Zhou, Yangfan Jia, Haiyang Tang

**Affiliations:** 1College of Veterinary Medicine, Northwest A&F University, Xianyang 712100, China; hr172052@163.com (R.H.); zhouxia1336508@163.com (X.Z.); jiayangfan0425@163.com (Y.J.); 2State Key Laboratory of Respiratory Disease, National Clinical Research Center for Respiratory Disease, Guangzhou 510120, China; 3Guangdong Key Laboratory of Vascular Disease, Guangzhou Institute of Respiratory Health, The First Affiliated Hospital of Guangzhou Medical University, Guangzhou 510120, China

**Keywords:** pulmonary arterial hypertension, hypoxia, pulmonary artery smooth muscle cells, proteomic, phosphoproteomic

## Abstract

Abnormal proliferation of pulmonary artery smooth muscle cells (PASMCs) is one of the main causes of pulmonary vascular remodeling in pulmonary arterial hypertension (PAH). Hypoxia is an important factor related to PAH and can induce the excessive proliferation of PASMCs and inhibit apoptosis. To explore the possible mechanism of hypoxia-related PAH, human PASMCs are exposed to hypoxia for 24 h and tandem mass tag (TMT)-based quantitative proteomic and phosphoproteomic analyses are performed. Proteomic analysis revealed 134 proteins are significantly changed (*p* < 0.05, |log2 (fold change)| > log2 [1.1]), of which 48 proteins are upregulated and 86 are downregulated. Some of the changed proteins are verified by using qRT-PCR and Western blotting. Phosphoproteomic analysis identified 404 significantly changed (*p* < 0.05, |log2 (fold change)| > log2 [1.1]) phosphopeptides. Among them, 146 peptides are upregulated while 258 ones are downregulated. The kinase-substrate enrichment analysis revealed kinases such as P21 protein-activated kinase 1/2/4 (PAK1/2/4), protein-kinase cGMP-dependent 1 and 2 (PRKG1/2), and mitogen-activated protein-kinase 4/6/7 (MAP2K4/6/7) are significantly enriched and activated. For all the significantly changed proteins or phosphoproteins, a comprehensive pathway analysis is performed. In general, this study furthers our understanding of the mechanism of hypoxia-induced PAH.

## 1. Introduction

Pulmonary arterial hypertension (PAH) is a fatal disease affecting not only the pulmonary artery system but also the function of the right ventricular. As a rare disease, the prevalence of PAH is from 15 to 50 per million in the United States and Europe [1]. Pulmonary vessel remodeling is an important characteristic of PAH and is often tightly related to the excessive proliferation and resistance of apoptosis of pulmonary arterial smooth muscle cells. Consequently, the lumen of pulmonary arterials becomes obstructed, and the blood pressure gets increased.

Clinically, pulmonary hypertension is classified into five groups, with which hypoxia is tightly associated with Group 3 [2]. Hypoxia is an important factor leading to pulmonary vascular remodeling and vasoconstriction, hence rodents exposed to chronic hypoxia are popularly used as animal models to study PAH [3]. In mammalian and human cells, the response to hypoxia is mainly mediated by hypoxia-induced factors (HIFs), such as HIF1α, HIF2α, and HIF1β. Under normal conditions, due to the high level of oxygen, specific proline residuals in HIF1α and HIF2α proteins are hydroxylated by prolyl hydroxylases (PHDs) family proteins PHD1, PHD2, and PHD3. Hydroxylated HIF1α and HIF2α are recognized and ubiquitinated by von Hippel-Lindau (VHL)/cullin2/elongin B ubiquitin E3 ligase complex and finally degraded by the proteasome [4]. However, under hypoxia conditions, the hydroxylation process is inhibited and hence HIF1α and HIF2α are stabilized and accumulated. Both HIF1α and HIF2α can dimerize with HIF1β and then enter the nucleus to drive the transcription of target genes with the help of other transcriptional factors. HIFs target genes affect the progression of PAH by regulating many important physiological processes such as vascular tone, angiogenesis, metabolism, and cell survival [5]. Especially in pulmonary artery smooth muscle cells (PASMCs), hypoxia-induced cell proliferation was related to the expression of miR-9-1 and miR-9-3, which were activated by HIF1α [6]. HIFs are highly PAH-relevant factors. The expression of both HIF1α and HIF2α has been found to be increased in PAH patients’ lung tissues [7,8]. In hypoxia-induced PAH mice models, suppression of the HIF2α signaling pathway either by gene knockout or HIF2α inhibitor protected the mice against PAH initiation and progression [9,10].

Protein phosphorylation is one of the most popular and important post-translational modifications. In PAH, phosphorylation of many proteins affects the progression of the disease, such as endothelial NOS (eNOS), forkhead box O 1 (FoxO1), ACE2 (angiotensin-converting enzyme 2), and AKT1 [11,12,13,14]. Systematically analyzing proteins and phosphorylated proteins involved in PAH helps to understand the underlying mechanisms in a broader view than studies focusing on single proteins. To globally analyze the proteins and phosphorylated proteins associated with hypoxia-induced PAH, here, an integrative proteomic and phosphoproteomic study was performed with hypoxia-treated human pulmonary artery smooth muscle cells (hPASMCs). For the differentially expressed proteins or phosphoproteins, we conducted a comprehensive signaling pathway and protein-protein interactions analysis. In addition, for proteomic analysis, the expressions of representative proteins were verified, and for phosphoproteomic analysis, kinase-substrate enrichment analysis (KSEA) was conducted to identify the activated or inhibited kinase.

## 2. Materials and Methods

### 2.1. Cell Culture and Treatment

Human pulmonary arterial smooth muscle cells (hPASMCs) were ordered from ScienCell Research Labs, and cultured in Smooth Muscle Cell Medium (Catalog#1101, ScienCell, Carlsbad, CA, USA). For normal cell culture, cells were maintained in a humidity incubator at 37 °C, 5% CO_2_. For hypoxia treatment, cells were maintained in similar conditions except that the incubator was additionally supplied with nitrogen and equipped with oxygen monitor to keep the O_2_ concentration at 1%.

### 2.2. Mass Spectrometry (MS) Samples Preparation

At 24 hrs before hypoxia treatment, hPASMCs were seeded into six 10 cm dishes. Before hypoxia treatment, the cell confluency was about 50–60%. The cells were maintained at normal conditions (normoxia) or hypoxia conditions (1% O_2_) for 24 hrs. After treatment, the cells were collected in 15 mL tubes with PBS buffer and then centrifugated at 200 g for 5 min. The cell pellets were lysed in 9 M urea (prepared in 50 mM HEPES, pH 8.5) supplemented with protease and phosphatase inhibitor (Pierce, Waltham, MA, USA), and transferred into Eppendorf tubes. The cell lysates were kept in ice for 30 min and vortexed every 5 min, followed by centrifugation at 4 °C, 14,000 g, for 20 min. Next, the supernatant was transferred into another tube and the protein concentration was determined with a BCA assay kit (Takara, Dalian, China). For each sample, 500 g total protein was subjected to reduction with 5 mM DTT and alkylation with 10 mM iodoacetamide. Then, the proteins were precipitated in methanol/chloroform, and the pellets were reconstituted in 1 M urea/50 mM HEPES (pH8.5). Overnight protein digestion was performed with sequencing-grade trypsin (Promega, Madison, WI, USA) at enzyme/protein ratio of 1:50, room temperature. Digested peptides were desalted with Oasis HLB 1 cc Vac Cartridge (30 mg, Waters, Milford, MA, USA) and dried to completeness in a vacuum-connected centrifuge.

Desalted peptides were reconstituted in 50 mM HEPES (pH8.5) and labeled with TMT10plex Mass Tag Labeling Reagents kit (90110, Themo Fisher Scientific, Shanghai, China). The following six channels were used: 126, 127C, and 128N for normoxia samples; 130N, 130C, and 131 for hypoxia samples. The labeling reaction was performed at room temperature for 1 h and quenched with hydroxyammine (Sigma, Shanghai, China). Then all the samples were combined into peptides mixture. For proteome analysis, 5% of the TMT-labeled peptides mixture was desalted with 1 cc Vac Cartridge and then fractionated with a High pH Reversed-Phase Peptide Fractionation Kit (Pierce) into 9 fractions according to the manufacturer’s instructions, the product of which was dried and then dissolved in 0.1% formic acid (FA) and submitted to LC-MS/MS analysis. For phosphopeptides enrichment, the rest of the peptide mixture was desalted with 3 cc Sep-Pak tC18 Vac Cartridge (200 mg; WAT054925, Waters).

After desalting, the peptide mixture was enriched with the High-Select Fe-NTA Phosphopeptide Enrichment Kit (A32992, Thermo Fisher Scientific, Shanghai, China). The eluted peptides were fractionated as above mentioned, while the flow-through peptides were further enriched with the High-Select™TiO_2_ Phosphopeptide Enrichment Kit (A32993, Thermo Fisher Scientific, Shanghai, China). The workflow for cell treatment and sample preparation is shown in Figure 1.

### 2.3. LC-MS/MS Analysis 

Peptides fractions for whole proteome analysis or phosphorylation analysis were reconstituted in 0.1% FA and submitted to Orbitrap Fusion Lumos Tribrid Mass Spectrometer connected with an Easy-nLC-1200 high-pressure liquid chromatography system (Thermo Fisher Scientific, Waltham, MA, USA). Each sample was loaded on a 15 cm C18 column (HS-Anal-C-5U-15CM, Beijing Happy Science Scientific, Beijing, China) of 5 μm mean particle size and 120 A pore diameter. Peptides were separated with a 80 min gradient at the flow rate of 300 nL/ min. For mobile phase, Buffer A is 0.1% FA, Buffer B is 0.1% FA/90% acetonitrile. MS1 spectra were collected with Orbitrap detector using these parameters: resolution of 120,000, scan range of 350-1800 (*m*/*z*), AGC target value of 4 × 10^5^, 50 ms maximum injection time (MIT), including charge states 2–6. MS2 spectra were collected in data-dependent mode, in which top 12 precursors were scanned, and the following parameters included: quadrupole isolation mode, 38% high energy collision dissociation (HCD) energy, auto scan range mode, 50,000 orbitrap resolution, 1 × 10^5^ AGC target value.

### 2.4. Database Searching 

MS raw files were searched against UniProtKB reference human proteome database (UP000005640), which contains 20,361 reviewed entries using Proteome Discoverer software 2.2 (Thermo Fisher Scientific). A template processing workflow (PWF_Fusion_Reporter_Based_ Quan_MS2_SequestHT_Percolator) within the software was used. Main searching parameters were set as follows: digestion enzyme: trypsin (full), maximal missed cleavage: 3, minimal peptide length: 6, precursor mass tolerance: 10 ppm, fragment mass tolerance: 6 Da, static modifications: cysteine residues carbamidomethylation (+57.021 Da) and N-terminus and lysine residue TMT-6plex (+229.163 Da), dynamic modifications included N-terminus acetylation (+42.011 Da), methionine residues oxidation (+15.995 Da). For phosphoproteome data processing, additional dynamic phospho/+79.966 Da modifications on serine, threonine, and tyrosine residues were added.

### 2.5. RNA Isolation and Quantitative Real Time PCR (qRT-PCR)

Total RNA was isolated from PASMCs by using RNAiso Plus (Takara, Beijing, China). In total, 1 μg of RNA was reverse-transcribed into 20 μL cDNA with Evo M-MLV RT Kit with gDNA Clean for qPCR II (AG, Changsha, China). The reverse transcription product was diluted by 1:14, and 3 ul diluent was taken for each qRT-PCR reaction. Quantitative real-time PCR was performed with 2× qPCR SmArt Mix (SYBR Green) (DIYIBIO, Shanghai, China), in a 20 uL reaction system. qRT-PCR was performed on CFX connect (Bio-Rad, Shanghai, China). Relative expression level of each gene was calculated by 2-ΔΔ method, in which the expression of Peptidylprolyl isomerase A (PPIA) was used as the internal control. Primers for qRT-PCR used in current study included the following: 

*ANGPT4L*: 5′-CCTCTCCGTACCCTTCTCCA-3′ (forward) and 5′-AAACCACCAGCCTCCAGAGA-3′(reverse); *ATP6AP1*: 5′-AGCGACTTGCAGCTCTCTAC-3′(forward) and 5′-CCTCAATGCTCAGCTTGTCC-3′(reverse); *CA12*: 5′-CAGTTTTCCGAAACCCCGTG-3′(forward) and 5′-GCAGTACAGACTTGCACTTGG-3′(reverse); *PPIA*: 5′-ATGGCGGTGGCAAATTCAAG-3′(forward) and 5′-CCGTCTTAGGCACAACGTCTG-3′(reverse).

### 2.6. Western Blotting Analysis

The cell lysate was prepared in ice-cold RIPA buffer (P0013C, Beyotime Biotechnol ogy, Shanghai, China), in which cocktail protease inhibitor (Thermo Fisher Scientific, Shanghai, China) was freshly added. The proteins were separated in homemade SDS-PAGE gel and transferred onto PVDF membrane. Blocking was performed with 5% non-fat milk (P0216, Beyotime Biotechnology, Shanghai, China) in TBST (Tris-buffered saline + 0.1% Tween 20). After blocking, the membrane was incubated with primary antibody overnight and washed with TBST four times, and then incubated with HRP-linked secondary antibody (Beyotime Biotechnology, Shanghai, China) for 2–3 h and washed with TBST. The signal was developed with WesternBright ECL HRP substrate (Advansta, Menlo Park, CA, USA) and captured using a chemiluminescent imaging system (Tanon 5200). Primary antibodies used in this study included HIF1a (ab179483, Abcam, Cambridge, UK), ANGPTL4 (18374-1-AP, Proteintech, Wuhan, China), ATP6AP1 (15305-1-AP, Proteintech, Wuhan, China), CA12 (15180-1-AP, Proteintech, Wuhan, China), and a-Tubulin (T6199, Sigma, Shanghai, China).

### 2.7. Statistical and Bioinformatics Analysis

For both proteins and phosphopeptides quantification, the sum of all the Scaled Abundance values in each TMT-channel was used to normalize peptides loading across different channels. Hence, for each protein or peptide, its normalized Scaled Abundance was regarded as its expression level and used for quantification and statistical analysis. For volcano plots, data were processed in Perseus 1.6.15.0 and the exported matrix were visualized with ggplot2 package in R Studio. Heatmaps were generated by using the Pheatmap function in R. Pathway analysis was performed online with (https://metascape.org, accessed on 12 May 2022) [15]. For phosphopeptides corresponding to more than one protein, each protein was used as an input when doing pathway enrichment analysis. Kinase-Substrate Enrichment Analysis was performed with the KSEA App [16]. Protein-protein interaction networks were analyzed online in String (https://cn.string-db.org/, accessed on 12 May 2022) and visualized with Cytoscape 3.8.2 [17].

## 3. Results

### 3.1. Proteomic and Phosphoproteomic Profiling of hPASMCs Exposed to Hypoxia

In the proteome study, a total of 3694 proteins were identified, of which 3526 proteins were quantified (Appendix A). Statistical analysis revealed 134 significantly changed proteins (*p* < 0.05, |log2 (fold change)| > log2 (1.1)), 86 downregulated proteins and 48 upregulated proteins (Appendix A, Figure 2A and Figure 3A). In phosphoproteome analysis, we identified a total of 9642 peptides, of which 6102 peptides were phosphorylated and quantified, which corresponded to 2347 phosphoproteins (Appendix A). Statistical analysis revealed 258 downregulated phosphopeptides and 146 upregulated phosphopeptides, respectively, corresponding to 219 and 142 phosphoproteins (Appendix A, Figure 2B and Figure 3C). To check the overlap between proteome analysis and phosphoproteome analysis, Venn diagrams were drawn. It showed that 1116 proteins were quantified in both proteome analysis and phosphoproteome analysis. In total, two proteins were upregulated and seven proteins were downregulated in both proteome analysis and phosphoproteome analysis (Appendix A).

### 3.2. Pathways Analysis of Significantly Changed Proteins and Phosphoproteins

To explore the potential pathways in which these significantly changed proteins and phosphoproteins are enriched, we performed a comprehensive pathway enrichment analysis by using a web-based portal called Metascape. Metascape integrates multiple commonly used databases for gene annotation, such as the Gene Ontology database, Kyoto Encyclopedia of Genes and Genomes (KEGG) database, Reactome database, and Molecular Signature database. It showed that the upregulated proteins are enriched in pathways such as carbon metabolism, regulation of transforming growth factor beta (TGF-β) receptor signaling pathway, and notably response to oxygen levels, which includes proteins such as ATPase H+ Transporting Accessory Protein 1 (ATP6AP1), Hexokinase 2 (HK2), Angiopoietin-like 4 (ANGPTL4), and solute carrier family 2 member 1 (SLC2A1) (Figure 3B and Appendix A). Downregulated proteins are enriched in pathways such as collagen, mRNA processing, vesicle-mediated transport, and diabetic cardiomyopathy (Figure 3B and Appendix A). For significantly upregulated phosphoproteins, representative enriched pathways are related to signaling by Rho GTPases, actin cytoskeleton organization, the VEGFA-VEGFR2 signaling pathway, and regulation of cellular localization (Figure 3D). Downregulated phosphoproteins are enriched in pathways such as the regulation of mRNA metabolic processes, Rho GTPase, chromatin organization, and nuclear export (Figure 3D and Appendix A).

### 3.3. Interaction Network for Proteins Changed in Proteome Analysis

To further elucidate the relationships of these proteins changed significantly under hypoxia, a protein-protein interaction (PPI) network was created (Figure 4A). In addition, as support to the pathway enrichment analysis, subnetworks were created for upregulated proteins enriched in the selected pathways, including carbon metabolism, regulation of the TGF-β receptor signaling pathway, and response to oxygen levels (Figure 4B–D).

### 3.4. Kinase-Substrate Enrichment Analysis for Hypoxia-Regulated Phosphoproteins in hPASMCs

To reveal the kinases whose activities were significantly changed by hypoxia treatment in hPASMCs, a kinase-substrate enrichment analysis (KSEA) was performed with all the quantified phosphopeptides (Figure 5A). It showed that hypoxia tended to inhibit the activities of kinases such as casein kinase 1 and 2 (CSNK2A1/2) and cyclin-dependent kinase 5 (CDK5) but activate P21 protein (Cdc42/Rac)-activated kinase 1/2/4 (PAK1/2/4), protein kinase cGMP-Dependent 1 and 2 (PRKG1/2) and mitogen-activated protein kinase 4/6/7 (MAP2K4/6/7). In addition, a kinase-substrate interaction network was created for the main enriched kinases and the potential phosphosites (Figure 5B).

### 3.5. qRT-PCR Verification of Selected Proteins

Finally, to verify the significantly changed proteins identified in proteomic analysis, mRNA and protein levels of representative proteins including ANGPTL4, ATP6AP1, and carbonic anhydrase 12 (CA12) were determined by using qRT-PCR (Figure 6A) and Western blotting (Figure 6B). It showed that hypoxia treatment significantly increased the expression of ANGPTL4, ATP6AP1, and CA12 both at the mRNA level and protein level in hPASMCs.

## 4. Discussion

In the PAH field, LC-MS/MS has been widely used to explore the differentially expressed proteins in the lung tissue of PAH patients compared with healthy control samples or in animal model lung tissues and control animals, while studies specific to the changed proteins in PASMCs during the progression of PAH are rare [18,19,20]. In the present study, we analyzed the proteome and phosphoproteome in hypoxia-treated hPASMCs by using a TMT-based quantitative proteomic strategy, hoping to provide some hints for the mechanism of hypoxia-related PAH. 

In proteome analysis, in total, we identified 134 significantly changed proteins in hPASMCs exposed to hypoxia. To check the overlap between the significantly changed proteins we identified and the findings of previous publications, we compared our protein list with the protein lists from two studies by Wu et. al. [20] and Xu et. al. [19] but found very low overlap (data not shown). The reason may be that we used primary hPASMCs while the others used either PAH patient lung tissues or isolated human pulmonary arterial endothelial cells. Of these 134 proteins, 48 were upregulated and 86 were downregulated. The expression of some of these upregulated proteins has been reported to be induced by hypoxia in other cell types, such as ANGPTL4 and CA12, the expressions of which were verified by using qRT-PCR and Western blotting. ANGPTL4 is a secreted protein, which is originally identified as a fasting-inducted adipose factor in mice liver and has been proved to be activated by hypoxia in endothelial cells and cardiomyocytes [21,22,23]. Importantly, ANGPTL4 also participates in inflammation, angiogenesis, and tumorigenesis, such as prostate cancer and hepatocellular carcinoma [24,25]. Considering the similarities in excessive cell proliferation between human cancer cells and PASMCs of PAH, it is very likely that ANGPTL4 plays some roles in PAH. CA12 is a member of the carbonic anhydrase family, which catalyzes the reversible hydration of carbon dioxide and bicarbonate and hence plays important roles in regulating intracellular pH [26]. It has been reported that the expression of CA12 has increased in human colon adenocarcinoma cells in a HIF1α-dependent manner and is necessary for tumor cell proliferation both in vivo and in vitro [27]. The latest study found that CA12 mediated the prometastatic functions of human hepatocellular carcinoma-associated macrophages [28]. While unspecifically inhibiting carbonic anhydrase ameliorates inflammation and experimental pulmonary hypertension and carbonic anhydrase IX plays important roles in regulating pulmonary microvascular endothelial cell pH and angiogenesis during acidosis, the functions of CA12 in PAH have not been studied yet [29,30]. 

For these changed proteins in hypoxia-treated PASMCs, pathway enrichment analysis was applied. One of the most significantly enriched pathways for upregulated proteins is carbon metabolism, and the relevant representative proteins include Hexokinase 2 (HK2), enolase 2 (ENO2), and SLC2A1, which is also called glucose transporter 1. This is in agreement with the conclusions of many previous studies that have proved glycolysis is enhanced in both PASMCs and pulmonary artery endothelial cells (PAECs) in PAH [31]. Another pathway enriched by upregulated proteins is the regulation of the TGF-β receptor signaling pathway, and the representative protein is Endoglin (ENG), which is a glycoprotein belonging to the TGF-β receptor system and has been proved to be elevated in the serum of systemic sclerosis patients with elevated systolic pulmonary arterial pressure [32,33].

In our phosphoproteome analysis, we totally identified 404 significantly changed (*p* < 0.05, |log2 (fold change)| > log2 [1.1]) phosphopeptides. To check if there are any novel phosphosites identified, significantly changed phosphoproteins containing precisely localized phosphosites were searched against the UniProtKB database (www.uniprot.org, accessed on 12 May 2022) and Phosphosite database (www.phosphosite.org, accessed on 12 May 2022). We found most of these phosphosites were known ones except L1RE1 (LINE-1 retrotransposable element ORF1 protein)-S18, CUL3 (Cullin-3)-S737, ERBIN-S1015, and MACF1 (Microtubule-actin cross-linking factor 1)-T5327 (Appendix A). In contrast to proteome studies, phosphoproteome analysis in PAH is relatively rare. A recent relevant study performed proteomics and phosphoproteomics analysis in PAH patient-derived PAECs [19], in which fewer significantly changed phosphopeptides were identified than we performed (240 VS. 404). One of the common pathways enriched by significantly changed phosphoproteins in this study and ours is signaling by Rho GTPases, which play important roles in cell motility and smooth muscle contractility by regulating actin dynamics, and have been proved to be implicated in several pulmonary vascular diseases including PAH [34]. Interestingly, we found the hypoxia-upregulated phosphoproteins were significantly enriched in the VEGFA-VEGFR2 signaling pathway and the EGF/EGFR signaling pathway, which have been proven to be tightly associated with the pathogenesis of PAH [35,36].

The best materials to study proteins involved in the progression of PAH are PASMCs and PAECs directly derived from the lung tissue of patients, which is a drawback of the present study. Another point that can be improved for our study is the use of a deeper fractionation method for LC-MS/MS sample preparation. In this way, more hypoxia-regulated proteins and phosphopeptides could be identified. 

## 5. Conclusions

Using TMT-based quantitative proteomic and phosphoproteomic, the present study globally analyzed the hypoxia-regulated proteins and phosphoproteins in hPASMCs, hoping to provide some mechanism hints for hypoxia-related PAH. In total, we identified 134 significantly changed proteins and 331 significantly changed phosphoproteins. The expression of changed proteins such as ANGPTL4, CA12, and ATP6AP1 was verified by using Western blotting. Pathway enrichment analysis revealed that these changed proteins are related to pathways such as carbon metabolism, regulation of TGF-β receptor signaling pathway, and response to oxygen levels, while changed phosphoproteins are related to Rho GTPases, actin cytoskeleton organization, VEGFA-VEGFR2 signaling pathway and regulation of cellular localization. In general, the findings of this preliminary study contribute to the understanding of hypoxia-related cell processes and hypoxia-induced PAH.

## Figures and Tables

**Figure 1 proteomes-10-00023-f001:**
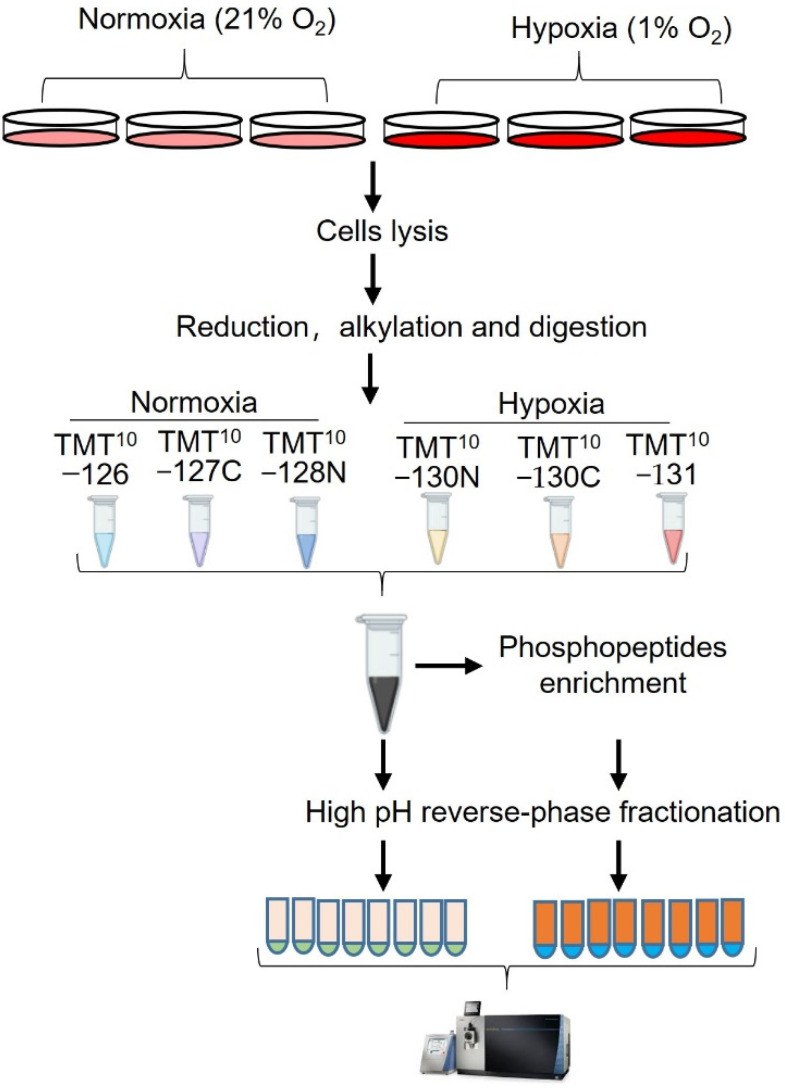
Experimental design and workflow for the study. hPASMCs were exposed to hypoxia (1%) for 24 h and the control cells were maintained at normal condition (21%). After treatment, the cells were lysed under denatured condition (8.5M urea/HEPES), and then reduced, alkylated, digested, and desalted. The desalted peptides were labeled with 6 channels of a TMT-10 plex kit. For proteome analysis, the 5% of the TMT−labeled peptides mixture was desalted and fractionated. For phosphoproteome analysis, the rest of the TMT−labeled peptides mixture was desalted and sequentially enriched with Fe-NTA and TiO_2_ column. Eluates of Fe-NTA column was further fractionated into 8 fractions and subjected to LC-MS/MS analysis together with eluate from TiO_2_ column.

**Figure 2 proteomes-10-00023-f002:**
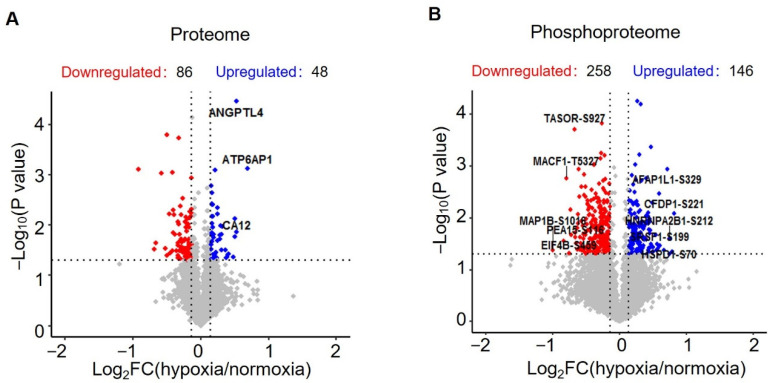
Statistical analysis of the quantified proteins and phosphopeptides. (**A**) Volcano plot showing the distribution of proteins quantified in proteomic analysis. (**B**) Volcano plot showing the distribution of phosphopeptides quantified in phosphoproteomic analysis.

**Figure 3 proteomes-10-00023-f003:**
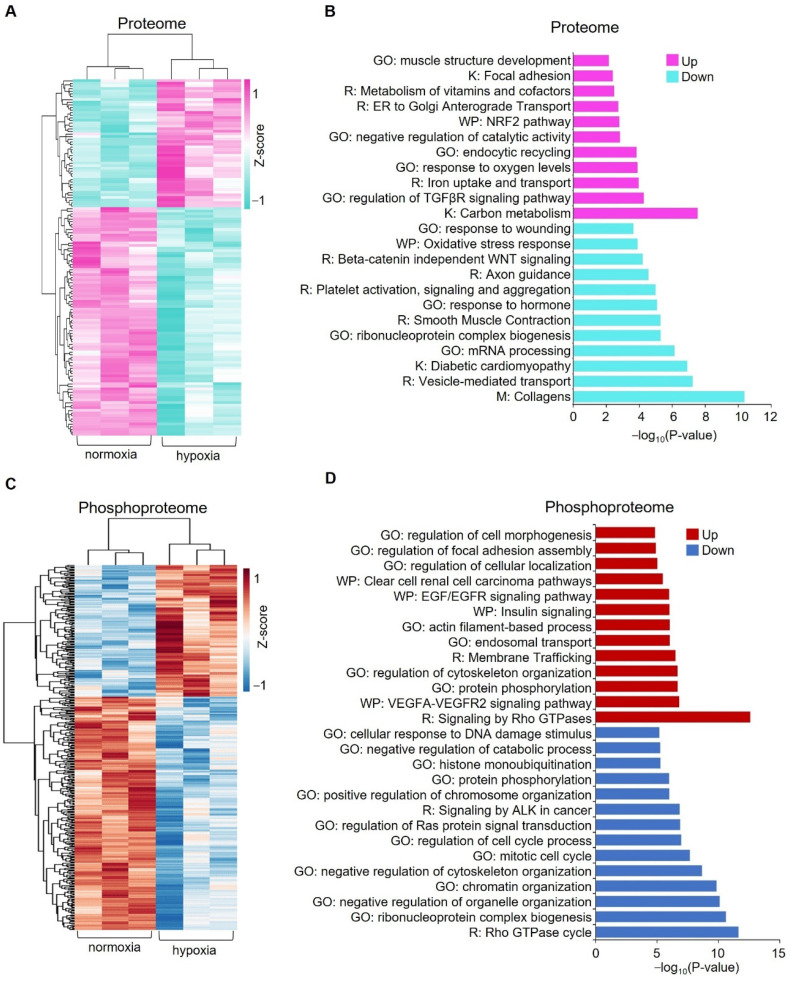
Pathways enriched for the significantly changed proteins and phosphoproteins between normoxia and hypoxia. (**A**) Heatmap showing the normalized abundances of significantly changed proteins (*p* < 0.05, |log2 (fold change)| > log2 (1.1)) in hypoxia-treated cells compared with the untreated cells (normoxia). (**B**) Pathway enrichment analysis of significantly changed proteins. Representative pathways are shown. (**C**) Heatmap showing the normalized abundances of significantly changed phosphopeptides (*p* < 0.05, |log2 (fold change)| > log2 (1.1)). (**D**) Pathway enrichment analysis of significantly changed phosphopeptides. Representative pathways are shown. GO: Gene ontology, K: KEGG database, R: Reactome database, WP: WikiPathways database.

**Figure 4 proteomes-10-00023-f004:**
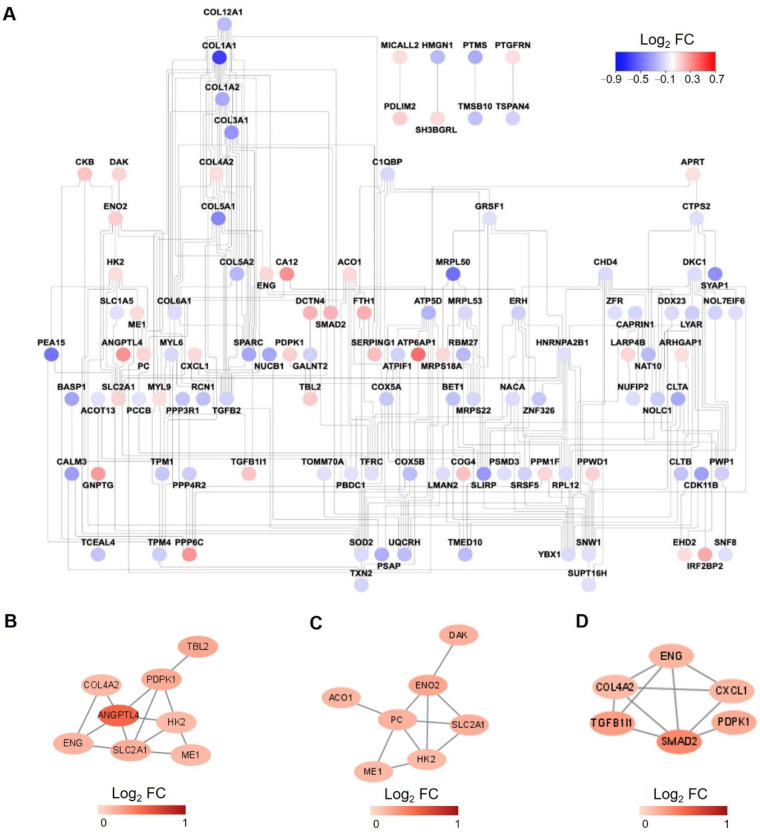
Protein-protein interaction (PPI) network analysis. (**A**) PPI network for all the significantly changed proteins in proteome analysis. The color of each node is assigned according to the log2 transformed fold change of the protein abundances of hypoxia to normoxia-treated samples. (**B**–**D**) PPI network for upregulated proteins enriched in carbon metabolism (**B**), response to oxygen levels (**C**), regulation of TGF-β receptor signaling pathway (**D**).

**Figure 5 proteomes-10-00023-f005:**
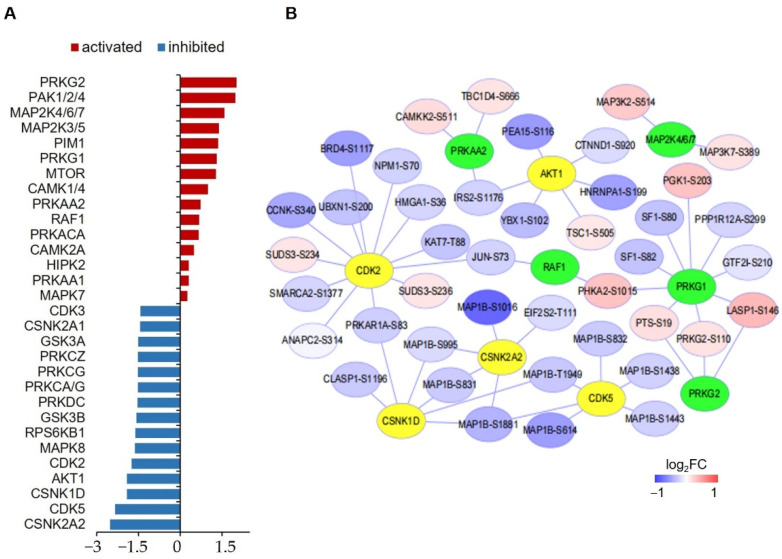
Kinase-substrate enrichment analysis of significantly changed phosphoproteins. (**A**) Bar graph showing the Z scores top-15 activated and inhibited kinases. (**B**) Interaction network between top-10 activated and inhibited kinases and the associated phosphoproteins. Kinases having only one substrate are not shown. Green nodes represent activated kinase, yellow nodes represent inhibited kinases.

**Figure 6 proteomes-10-00023-f006:**
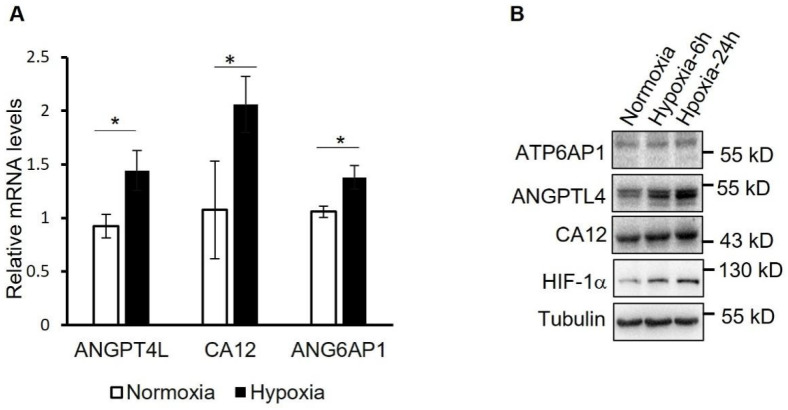
Verification of the proteins upregulated by hypoxia in PASMCs. (**A**) mRNA levels of the indicated proteins were determined by using qRT-PCR in PASMCs treated with hypoxia for 24 h or not. * *p* < 0.05, Student’s t test, *n* = 3. (**B**) Normoxia- or hypoxia-treated hPASMCs were analyzed by using Western blotting with the indicated antibodies.

## Data Availability

All the raw data are available upon reasonable request.

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
