# Peer review of "Integrative Proteomic and Phosphoproteomic Analyses of Hypoxia-Treated Pulmonary Artery Smooth Muscle Cells"

_proteomes, 2022, doi:10.3390/proteomes10030023_

Round 1

Reviewer 1 Report

Hao et. al. studied the proteomic and phosphoproteomic analysis of hypoxia-treated pulmonary artery smooth muscle cells. I have few issues with the manuscript.

  1. How is this study different from Xu, Weiling et al. “Integrative proteomics and phosphoproteomics in pulmonary arterial hypertension.” Scientific reports vol. 9,1 18623. 9 Dec. 2019, doi:10.1038/s41598-019-55053-6 (PMID: 31819116) in terms of applicability?
  2. Why the authors selected 24 hr exposure time?
  3. Experimental workflow figure 1a would be more suited in the methodology section instead of results section. Please revise.

Author Response

  • How is this study different from Xu, Weiling et al. “Integrative proteomics and phosphoproteomics in pulmonary arterial hypertension.” Scientific reports vol. 9,1 18623. 9 Dec. 2019, doi:10.1038/s41598-019-55053-6 (PMID: 31819116) in terms of applicability?

In Weiling Xu’s paper (“Integrative proteomics and phosphoproteomics in pulmonary arterial hypertension.” Scientific reports vol. 9,1 18623. 9 Dec. 2019, doi:10.1038/s41598-019-55053-6), the authors used pulmonary arterial endothelial cells (PAECs) from PAH patients as the material, but we used primary pulmonary arterial smooth muscle cells (PASMCs) from healthy donors. In the progression of PAH, PAECs and PASMCs functions differently. For example, as two of the most important hypoxia-induced factors, the expression of HIF1a is high in PASMCs but low in PAECs, while the expression of HIF2a is high in PAECs but low in PASMCs. In addition, findings from our study is not only useful to the field of PAH but also to the field of hypoxia-related biology.

  • Why the authors selected 24 hr exposure time?

We tested both 6 hr and 24 hr for the exposure time of hypoxia, and found 24 hr exposure time can induced the expression of HIF1a more efficiently than 6 hr. Hence, we selected 24 hr.

  • Experimental workflow figure 1a would be more suited in the methodology section instead of results section. Please revise.

Thanks for your advice. In the revised version of the manuscript, we moved Figure 1a to the Methods and Materials as Figure 1.

Reviewer 2 Report

The manuscript entitled “Integrative proteomic and phosphoproteomic analysis of hy-2 poxia-treated pulmonary artery smooth muscle cells 3” presented TMT-LC-MS/MS based proteome profiling of PASMCs under hypoxia perturbation. Together with functional enrichment analysis and validation, this work provide valuable resource in related field and insight into the mechanism of hypoxia-induced pulmonary arterial hypertension. It is recommended to publish this work if the following comment can be responded, and suggestions can be addressed in revised manuscript.

1)      In section 2.2, please specify the cell confluency or cell number plated before perturbation.

2)      In line 84, there is a typo – “500 g”. I think the author was using 500 ug proteins as input rather than gram scale.

3)      In section 2.2, was the Fe-NTA enriched sample and TiO2 enriched sample combined for analysis or were analyzed separately?

4)       In section 2.4, how did the author perform the peptide/protein filtering and FDR control?

5)      In line 173, how was log2(1.1) determined? What is the overall SD of log2FC?

6)      In fig 1b and 1c, the name of y-axis should be “-Log10(P value)” and the name of x-axis should be “Log2FC(hypoxia/normoxia)”. And it maybe better to annotate the protein of importance, e.g. proteins discussed in main text, on the volcano plot.

7)      Regarding the small changes between hypoxia and normoxia, I wonder how is PCA analysis of replicates?

8)      In fig 2a and 2c, it is better to put a subtitle, e.g. proteome, phosphoproteome, to differentiate two heatmap rather than only use different color scale. And please specify the color scale in the figure, e.g. fold change or TMT reporter intensity.

9)      In fig 2b and 2d, please add legend for colors, e.g. up-regulated and down-regulated. Some pathway names are all upper cases, Please universalize the format of those names.

10)   In fig 2b and 2d, many of those pathways haven't been discussed in main text. Please make those two figures more focused on pathway of most importance. Comprehensive figures of less important pathways can be shown in supplementary information.

11)   In fig 3b-d, could the author also indicate the up-regulation of down-regulation of those proteins in each network?

12)   In fig 4a, please add legend for colors, e.g. activated and inhibited.

13)   In fig 4b, did those kinases of yellow and green node have any protein level change?

14)   In fig 5, did the author validated some down-regulated proteins or used other protein-readout assay like western blot for validation experiments?

Author Response

Reviewee 2#

  • In section 2.2, please specify the cell confluency or cell number plated before perturbation.

In section 2.2 of the revised version of the manuscript, we added “Before hypoxia treatment, the cell confluency was about 50%-60%.”

  • In line 84, there is a typo – “500 g”. I think the author was using 500 ug proteins as input rather than gram scale.

Thank you very much for pointing out this mistake. Yes, we used 500 ug proteins as input, and we corrected the typing mistake.

  • In section 2.2, was the Fe-NTA enriched sample and TiO2 enriched sample combined for analysis or were analyzed separately?

Fe-NTA enriched sample and TiO2 enriched sample were loaded separately on the LC-MS/MS system, but the raw files were analyzed together as different fractions in the Proteome Discoverer software.

  • In section 2.4, how did the author perform the peptide/protein filtering and FDR control?

Peptide and protein filters include:Peptide Confidence At Least was set at “High”, Keep Lower Confident PSMs was set at “False”, Minimum Peptide Length was set at 6, Minimum Number of Peptide Sequences was set at 1, Count Only Rank 1 Peptides was set at “False”. Protein FDR Validator was set at “Target FDR=0.01”.

5)      In line 173, how was log2(1.1) determined? What is the overall SD of log2FC?

Log2(1.1)was chosen arbitrarily to include as much proteins as possible for analysis. The overall SD of log2FC is 0.24.

6)      In fig 1b and 1c, the name of y-axis should be “-Log10(P value)” and the name of x-axis should be “Log2FC(hypoxia/normoxia)”. And it maybe better to annotate the protein of importance, e.g. proteins discussed in main text, on the volcano plot.

Thank you much for pointing out these errors. In the new version of the manuscript, we made the relevant changes for the original Figure 1b and 1c, which are now Figure 2A and 2B, and representative proteins are annotated on the volcano plots.

7)      Regarding the small changes between hypoxia and normoxia, I wonder how is PCA analysis of replicates?

We performed PCA analysis for the proteome analysis data and put the result in Supplementary Figure S1.

8)      In fig 2a and 2c, it is better to put a subtitle, e.g. proteome, phosphoproteome, to differentiate two heatmap rather than only use different color scale. And please specify the color scale in the figure, e.g. fold change or TMT reporter intensity.

Thank you very much for your advice, in the new version of the manuscript we made the relevant changes to the original Figure 2a and 2c, which are now Figure 3A and 3C.

9)      In fig 2b and 2d, please add legend for colors, e.g. up-regulated and down-regulated. Some pathway names are all upper cases, Please universalize the format of those names.

Thank you for your advice. In the revised version of fig2b and 2d, which are now Figure 3B and 3D, format of all the pathway names is universalized.

10)   In fig 2b and 2d, many of those pathways haven't been discussed in main text. Please make those two figures more focused on pathway of most importance. Comprehensive figures of less important pathways can be shown in supplementary information.

Thank you for your advice. In the new version of the manuscript, only representative pathways are included in Fig 2b and 2d, which are now Figure 3B and 3D, original Fig 2b and 2d was moved to Supplementary Figure S3 and S4.

11)   In fig 3b-d, could the author also indicate the up-regulation of down-regulation of those proteins in each network?

Thank you for your advice. In the revised version of fig3b-d, which are now Figure 4B-D, fold change values were assigned to these protein nodes in each network. Actually, all these proteins were upregulated.

12)   In fig 4a, please add legend for colors, e.g. activated and inhibited.

Thanks for pointing this error. In Fig 5A, which corresponds to original Fig 4a, activated and inhibited information was labeled in the figure.

13)   In fig 4b, did those kinases of yellow and green node have any protein level change?

In the revised version of the manuscript, the phosphoproteome data was reanalyzed, and hence we updated the KSEA analysis. In the new Figure 5B, most of these kinases of yellow and green nodes were not identified in our proteome analysis, except MAP2K4, PRKG2 and AKT1. But the protein levels of these identified kinases were not significantly changed under hypoxia.

14)   In fig 5, did the author validated some down-regulated proteins or used other protein-readout assay like western blot for validation experiments?

In the revised version of the manuscript,upregulated proteins shown in original Figure 5, which is now Figure 6A, was further validated by using western blot (Figure 6B). We are sorry that we did not validate any down-regulated proteins. Upregulated proteins under hypoxia are usually regulated by hypoxia-induced factors (HIFs), but mechanism regarding to the downregulated proteins is complicated.

Reviewer 3 Report

The manuscript by Hao et. al attempts a proteomic and phosphoproteomic approach to study Pulmonary arterial hypertension (PAH) using human pulmonary arterial smooth muscle cells (hPASMCs) cell lines. The cell lines were treated under hypoxic (1%) and normal conditions and their proteomic and phosphoproteomics profiles were compared using TMT-based approach. A total of 3694 proteins were identified in proteome and around 9600 phosphorylated peptides from the phosphoproteome data. The numbers are low with current mass spectrometry standards. The introduction is poorly written with mostly papers on non-human.

·      It would be nice if the authors could atleast compare the proteins with existing literature to show overlap. Also, how many of the phosphor sites are known and novel, some comparisons to existing databases is recommended.

·     It is not clear what “removing replicated data” means in phosphoproteome analysis. It would be nice to include the details of phospho analysis in methods and highlight some important proteins in the volcano plots. How was the phosphorylation data normalized? Was total protein taken into consideration?

·    Was there any overlap between the dysregulated proteins and dysregulated phosphopeptides results? It would be interesting to see if the changes are only at PTM level or the entire protein itself.

·   The authors did an enrichment using the Metascape tool. Was the total proteome used as background or the entire human proteins? If the later, then that would give wrong interpretations.

·      The authors attempted to validate mRNA levels of three proteins from the dysregulated proteins list. It is not clear how the three genes were chosen? Are these top differentials? What about phosphor data? It is well known that changes in proteome and phosphor levels are not often reflected in mRNA levels. mRNA and proteins shows poor correlation (0.2-0.4), please check any of the CPTAC papers (https://www.cell.com/consortium/cptac). Are these proteins known to be highly correlated in protein and mRNA levels?

·      The authors choose to validate three genes using RT-PCR. The proteomic evidence for 2 of them is low. Only 1 peptide/PSM for each protein. How do the authors justify the case and rationale to choose them?

·      The results are vague with very little hypothesis or possible discussion. The authors choose to describe three of the proteins in detail. What about the top enriched pathways and other proteins? What does the phosphoproteome analysis reveal?

·      Of the 3694 proteins identified in the current study, more than 1000 proteins are single peptide ones with even one or two PSMs. How do the authors justify their existence? Probably due to low fraction numbers? Of the differentially expressed proteins what is peptide support?

.   The authors should look at some publicly available resources like Human Protein Atlas (HPA) for expression of the proteins and their concordance.

Some other minor comments:

Typos:

Page 1 Line 1: Pulmonary arterial hypertension (PAH) is a “fetal” disease. Fatal?

Page 2 Line 1: nucleus and not “nuclear”

TGF R complex?

PAMCs in Page 2, first paragraph needs to be expanded in first use.

Similarly, PASMCs in Page 2 second paragraph

Section 2.1: “humiliated” incubator?

Author Response

Reviewee 3#

The manuscript by Hao et. al attempts a proteomic and phosphoproteomic approach to study Pulmonary arterial hypertension (PAH) using human pulmonary arterial smooth muscle cells (hPASMCs) cell lines. The cell lines were treated under hypoxic (1%) and normal conditions and their proteomic and phosphoproteomics profiles were compared using TMT-based approach. A total of 3694 proteins were identified in proteome and around 9600 phosphorylated peptides from the phosphoproteome data. The numbers are low with current mass spectrometry standards. The introduction is poorly written with mostly papers on non-human.

Thank you very much for your comments. The reason that the protein and peptide numbers are low is that our fractionation efficiency is low and we are trying to optimize the method in other project. The reason that we cited many papers on non-human in the introduction is that PAH is a rare disease and clinical samples are usually hard to get and hence animal models are widely used for basic research.

It would be nice if the authors could at least compare the proteins with existing literature to show overlap. Also, how many of the phosphor sites are known and novel, some comparisons to existing databases is recommended.

In the Discussion of revised manuscript version, from line 288, we added “To check overlap between the significantly changed proteins we identified and findings of previous publications, we compared our proteins list with protein list from two studies by Wu et.al and Xu et.al, but found very low overlap (data not shown). The reason may be that we used primary hPASMCs but the others used either PAH patient lung tissues or isolated human pulmonary arterial endothelial cells.” In line 324, we added “To check if there are any novel phosphosites identified, significantly changed phospoproteins containing precisely localized phosphosites were searched against the UniProtKB database (www.uniprot.org) and Phosphosite database (www.phosphosite.org). We found most of these phosphosites were known ones except L1RE1 (LINE-1 retrotransposable element ORF1 protein)-S18, CUL3 (Cullin-3)-S737, ERBIN-S1015, MACF1 (Microtubule-actin cross-linking factor 1)-T5327 (Supplementary Table S4).”

It is not clear what “removing replicated data” means in phosphoproteome analysis. It would be nice to include the details of phospho analysis in methods and highlight some important proteins in the volcano plots. How was the phosphorylation data normalized? Was total protein taken into consideration?

“Removing replicated data” means that only one peptide was randomly chosen for statical analysis when multiple peptides contained the same phosphosite. For data normalization, the sum of scaled abundances (reporter ion intensity) of each TMT channel was normalized to account for the differences in initial protein or peptide concentration. Then the normalized peptides abundances were used for further analysis. However, we found some mistakes in data normalization in the first manuscript version, hence we reanalyzed the phosphorylation data without removing the replicated data and all the relevant figures were updated in the revised manuscript version. Details of phosphoproteomic data analysis were described in 2.7. In addition, we added “For phosphopeptides correponding to more than one protein, each protein was used as an input when doing pathway enrichment analysis.” What’s more, representative phosphoproteins were highlighted in the volcano plot.  However, due to the low proteome coverage of our study, we did not take the total protein into consideration.

Was there any overlap between the dysregulated proteins and dysregulated phosphopeptides results? It would be interesting to see if the changes are only at PTM level or the entire protein itself.

In the revised version of the manuscript, in line 203 we added “To check the overlap between proteome analysis and phosphoproteome analysis, Venn diagrams were drawn. It showed that 1116 proteins were quantified in both proteome analysis and phosphoproteome analysis, 2 proteins were  upregulated and 7 proteins were downregulated in both proteome analysis and phosphoproteome analysis (Supplementary Figure S2).”

 The authors did an enrichment using the Metascape tool. Was the total proteome used as background or the entire human proteins? If the later, then that would give wrong interpretations.

For pathway enrichment analysis, the total proteome rather than the entire human proteins was used as the background.

The authors attempted to validate mRNA levels of three proteins from the dysregulated proteins list. It is not clear how the three genes were chosen? Are these top differentials? What about phosphor data? It is well known that changes in proteome and phosphor levels are not often reflected in mRNA levels. mRNA and proteins shows poor correlation (0.2-0.4), please check any of the CPTAC papers (https://www.cell.com/consortium/cptac). Are these proteins known to be highly correlated in protein and mRNA levels?

The reason that we choose these three genes is that they are the top-three upregulated proteins that we identified based on fold change. Phosphorylation sites were not found for these three proteins in our data. In addition, in the revised version of the manuscript,expression of these three proteins was additionally validated by using western blot (Figure 6B). The results showed that expression of these three proteins were increased both at mRNA and protein level in hPASMCs under hypoxia comparing to normoxia condition.

 The authors choose to validate three genes using RT-PCR. The proteomic evidence for 2 of them is low. Only 1 peptide/PSM for each protein. How do the authors justify the case and rationale to choose them?

For the three genes that we choose to validate, only 1 peptide/PSM was identified for ATP6AP1 and CA12, but ANGPT4L has 4 peptide/PSM. The reason that we choose these genes is that they are the top-three upregulated proteins that we identified based on fold change.

 The results are vague with very little hypothesis or possible discussion. The authors choose to describe three of the proteins in detail. What about the top enriched pathways and other proteins? What does the phosphoproteome analysis reveal?

In the Discussion, from line 314 to line 340, we described the top enriched pathways and other proteins, and discussed the results of phosphoproteome analysis. “One of the most significantly enriched pathways for upregulated proteins is carbon me-tabolism, and the relevant representative proteins include Hexokinase 2 (HK2), enolase 2 (ENO2) and SLC2A1 which is also called glucose transporter 1. This is in agree with the conclusions of many previous studies that have proved glycolysis is enhanced in both PASMCs and pulmonary artery endothelial cells (PAECs) in PAH [31]. Another pathway enriched by upregulated proteins is regulation of TGF-b receptor signaling pathway, and the representative protein is Endoglin (ENG), which is glycoprotein belong to the TGF-b receptor complex and has been proved to be elevated in the serum of systemic sclerosis patients with elevated systolic pulmonary arterial pressure [32].

In our phosphoproteome analysis, we totally identified 404 significantly changed (P < 0.05, |log2 (fold change) | > log2 [1.1]) phosphopeptides. To check if there are any novel phosphosites identified, significantly changed phospoproteins containing precisely lo-calized phosposites were searched against the UniProtKB database (www.uniprot.org) and Phosphosite database (www.phosphosite.org). We found most of these phosposites were known ones except L1RE1 (LINE-1 retrotransposable element ORF1 protein)-S18, CUL3 (Cullin-3)-S737, ERBIN-S1015, MACF1 (Microtubule-actin cross-linking factor 1)-T5327 (Supplementary Table S4). In contrast to proteome study, phosphoproteome analysis in PAH is relatively fewer. A recent relevant study performed proteomics and phosphoproteomics analysis in PAH patients-derived PAECs [19], in which fewer significantly changed phosphopeptides were identified than we did (240 VS 404).  One of the common pathways enriched by significantly changed phosphoproteins in this study and ours is signaling by Rho GTPases, which play important roles in cell motility and smooth muscle contractility by regulating actin dynamics, and have been proved to be implicated in several pulmonary vascular diseases including PAH [33]. Interestingly, we found the hypoxia-upregulated phosphoproteins were significantly enriched in VEGFA-VEGFR2 signaling pathway and EGF/EGFR signaling pathway, which has been proved to be tightly associated with the pathogenesis of PAH [34,35].”

Of the 3694 proteins identified in the current study, more than 1000 proteins are single peptide ones with even one or two PSMs. How do the authors justify their existence? Probably due to low fraction numbers? Of the differentially expressed proteins what is peptide support?

Yes, due to low fractionation depth or fraction number, the sequence coverage for more than 1000 proteins is very low, which makes these proteins less confident than other proteins with multiple peptides. However, in this study we used TMT-based quantification method, for protein with single peptide identified, it actually meant that this peptide was found in 6 samples, which were labeled with 6 different TMT reagent and mixed together for analysis. This is to say that it is very likely that these proteins were in the samples. Admittedly, additional verification experiment is necessary for specific protein to draw a conclusion. For the 134 differentially expressed proteins, totally 33 ones were identified with single peptide.

The authors should look at some publicly available resources like Human Protein Atlas (HPA) for expression of the proteins and their concordance.

Thank you for your advice. We checked the HPA database for the expression of ATP6AP1, CA12 and ANGPT4L in lung tissue, and found the levels of CA12 and ANGPTL4 is very low but ATP6AP1 showed medium expression level.

Some other minor comments:

Typos:

Page 1 Line 1: Pulmonary arterial hypertension (PAH) is a “fetal” disease. Fatal?

Thank you very much, this is a typing error. The right word is “fatal” and we have made the change.

Page 2 Line 1: nucleus and not “nuclear”

Thank you very much. “nuclear” is substituted by “nucleus” in the revised manuscript.

TGF R complex?

 We are sorry for the mistake. It’s “TGF-β receptor system” (line 321).

PAMCs in Page 2, first paragraph needs to be expanded in first use.

Similarly, PASMCs in Page 2 second paragraph

Thank you for point out this for us. In the revised version of the manuscript, we changed the relevant sentences according to your advice.

Section 2.1: “humiliated” incubator?

We are sorry for this error. The right word is humidity incubator. We have made the change in the revised version of the manuscript.

Round 2

Reviewer 1 Report

Authors have incorporated all the suggestions and comments given earlier. 

Reviewer 2 Report

The author of this manuscript have responded to all comment and revised the manuscript according to my suggestions. It is recommended to publish this work without any further revision.